# Hyperbolic material enhanced scattering nanoscopy for label-free super-resolution imaging

Yeon Ui Lee ®[1,4], Shilong Li[1,5], G. Bimananda M. Wisna[2], Junxiang Zhao[1], Yuan Zeng[2,3], Andrea R. Tao[2,3] & Zhaowei Liu ®[1,2] ✉

Fluorescence super-resolution microscopy has, over the last two decades, been extensively developed to access deep-subwavelength nanoscales optically. Label-free super-resolution technologies however have only achieved a slight improvement compared to the diffraction limit. In this context, we demonstrate a label-free imaging method, i.e., hyperbolic material enhanced scattering (HMES) nanoscopy, which breaks the diffraction limit by tailoring the light-matter interaction between the specimens and a hyperbolic material substrate. By exciting the highly confined evanescent hyperbolic polariton modes with dark-field detection, HMES nanoscopy successfully shows a high-contrast scattering image with a spatial resolution around 80 nm. Considering the wavelength at 532 nm and detection optics with a 0.6 numerical aperture (NA) objective lens, this value represents a 5.5-fold resolution improvement beyond the diffraction limit. HMES provides capabilities for super-resolution imaging where fluorescence is not available or challenging to apply.

Super-resolution microscopy, or nanoscopy, has become a critical tool for cell biology[1], neuroscience[2], pharmaceutical industry[3], and nanophotonic applications[4]. The widely used super-resolution nanoscopy techniques are mostly based on fluorescence[5–7], which have the capability to selectively label the targets on biological specimens. This fluorescent probe-based method provides exquisite image contrast resulting from high emission intensity contrast between the bright versus dark states of fluorophores; however, the restrictions in labeling protocols and alteration in cellular functions limit the range of imaging specimens[8]. Therefore, there is an urgent need to develop label-free super-resolution microscopy technologies.

Various label-free super-resolution imaging methods have recently been proposed[9]. Among the intensity-based detection methods, two techniques named interferometric scattering (iSCAT) microscopy[10,11] and single-particle interferometric reflectance (SPIR) microscopy[12] have a sensitivity sufficient to detect the position of nanoscopic objects, where the scattered field from the objects is superposed with a coherent reference field. In another approach, rotating coherent scattering (ROCS) microscopy[13–16] has been explored and showed how interferences can be used to improve image resolution and contrast by adopting an oblique, coherent illumination in dark-field mode.

Nevertheless, in contrast to fluorescence super-resolution nanoscopies, the resolution of the current super-resolution scattering imaging techniques is limited by the substrate materials that support a high effective refractive index at the desired wavelength. On one hand, dielectric materials have a limited refractive index. For example, super-resolution scattering image with a ~3-fold resolution improvement has been obtained by using GaP[17] and it is hard to find any other materials which are better than GaP. On the other hand, plasmonic materials have shown much higher effective refractive indices at visible frequencies; therefore, super-resolution fluorescence microscopies based on

[1]Department of Electrical and Computer Engineering, University of California, San Diego, 9500 Gilman Drive, La Jolla, CA 92093, USA. [2]Material Science and Engineering Program, University of California, San Diego, 9500 Gilman Drive, La Jolla, CA 92093, USA. [3]Department of NanoEngineering, University of California, San Diego, 9500 Gilman Drive, La Jolla, CA 92093, USA. [4]Present address: Department of Physics, Chungbuk National University, Cheongju, Chungbuk 28644, South Korea. [5]Present address: Light-Matter Interactions for Quantum Technologies Unit, Okinawa Institute of Science and Technology Graduate University, Onna, Okinawa 904-0495, Japan. ✉e-mail: zhaowei@ucsd.edu

plasmonic materials ranging from surface plasmon polariton structures[18–20] to hyperbolic metamaterials have recently been theoretically proposed and experimentally demonstrated[21–23]. However, these plasmonic materials have only been applied to fluorescence imaging based on our knowledge. As a result, current label-free scattering imaging techniques have only demonstrated a resolution improvement of ~3 fold[24–27].

In this work, we propose a label-free super-resolution imaging technique, named hyperbolic material enhanced scattering (HMES) nanoscopy, which will bring scattering images with a lateral resolution of 80 nm—a 5.5-fold resolution improvement over the diffraction limit —and beyond. The core working principle is to introduce a hyperbolic material as the substrate to not only enhance the scattering intensities of small objects but also control the illumination/scattered field distributions at deep subwavelength scales. The hyperbolic materials— anisotropic materials[28–31] which have permittivity of opposite sign along different directions—support highly directional propagation of volumetrically confined hyperbolic polaritons (high-spatial frequency, i.e., high-$k$ optical modes), which have been widely used for applications relying on light manipulation at sub-diffraction-limited length scales, e.g., biosensor[32], super-resolution imaging[33,34], and broadband enhancement of spontaneous emission[35]. Here, we use an organic hyperbolic material (OHM)[36–38] as the hyperbolic material because of its fabrication simplicity and additional flexibility in tailoring the dispersion of the hyperbolic polariton modes[36]. The scattering images of dielectric nanoparticles situated on top of the OHM were recorded with illumination beams from various incident angles with dark-field detection, similar to ROCS microscopy[13–16]. Since the recorded images come from the frequency mixing between the high-$k$ speckles from the substrate and the object, a super-resolved scattering image can be numerically retrieved via a blind structured illumination microscopy (Blind-SIM) reconstruction algorithm[39]. We observed that HMES nanoscopy resolves adjacent beads with a center-to-center resolution of 80 nm (NA = 0.6 at 532 nm).

## Results

### Principle of HMES nanoscopy

Figure 1a–c shows schematics of the proposed HMES nanoscopy. The setup is based on a common reflection microscope with slight modifications. A high-speed scanning mirror system, a custom radial polarizer (see Methods for details), and a focusing lens are introduced such that the incident azimuthal angle $\phi$ of a $p$-polarized laser beam ($\lambda = 532$ nm) can be swept from 0° to 360°. Meanwhile, the polar angle, $\theta$, is confined between 60° and 70° by focusing the laser beam on the edge of the back focal plane (BFP) of the objective lens. The scattering objects, i.e., polystyrene beads (Fig. 1d) are drop-casted on the top surface of the OHM[36], while a layer of random silver nanoparticles is attached on the other side (Fig. 1e). When the incident light hits on the sample, high-resolution electric field ($E$-field) hotspots are formed around the object due to the intricate interferences of the scattering light from both object and the silver nanoparticles assisted by the OHM. Part of the propagating components of the scattered light is collected by an imaging camera synchronized with the scanning mirror (frame rate of 10 frames per second). A diaphragm iris is used to block the directional reflection (surface-reflected light) in a plane conjugate to the BFP (see Fig. 1a and Methods for details) ensuring the dark-field detection. Only the scattered light from the polystyrene beads passes through and forms the scattering image. Because a different incident angle leads to a different hotspot distribution in the near field and thus a different scattering image in the far field, a series of scattering images are collected with various sweeping angles $\phi$ and $\theta$. A super-resolution image can be numerically reconstructed similar to the speckle structured illumination microscopy[39].

The OHM property plays a key role in the HMES nanoscopy. The horizontal $x$- and vertical $z$-components of the complex permittivity of regioregular poly(3-hexylthiophene-2,5-diyl) (rr-P3HT) OHM[36] are shown in Supplementary Fig. S1a, which were obtained from the variable angle spectroscopic ellipsometry (VASE) measurement[40] using anisotropic model[36,37] (see Methods for details). Such OHMs support

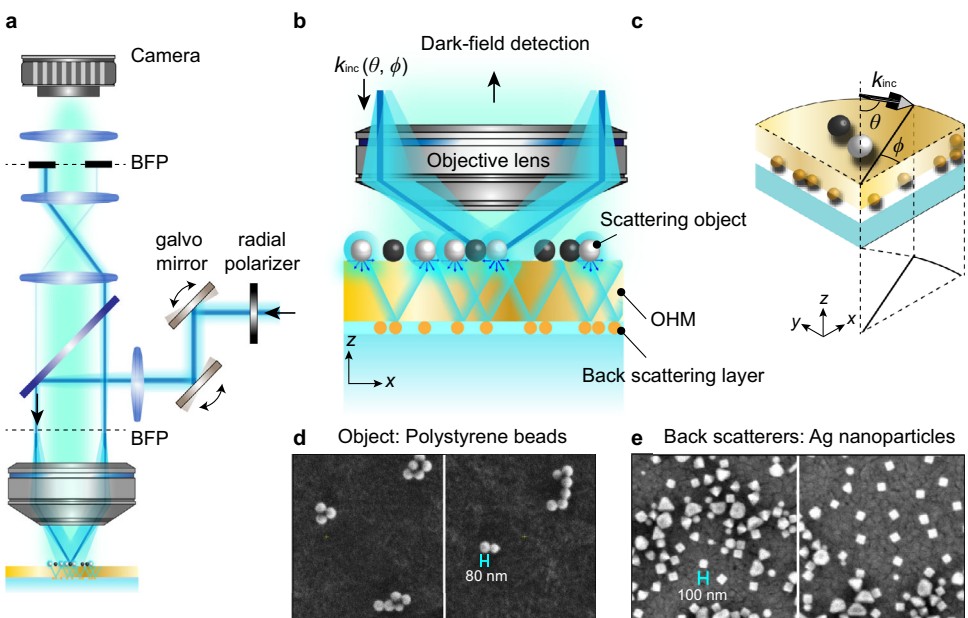

**Fig. 1 | Experimental setup of the HMES nanoscopy. a–c** Schematic of the HMES nanoscopy (**a**) and an enlarged view of it (**b**, **c**). An excitation laser beam ($\lambda = 532$ nm, $p$-polarized) was steered via scanning galvo mirrors through a 4$f$ system, by which the polar angle, $\theta$, and azimuthal angle, $\phi$, of the incidence beam were defined. The scattered light from the object was passed through a diaphragm at back focal plane (BFP) and then collected by a sCMOS camera. **d**, **e** Scanning electron microscope (SEM) images of test objects, i.e., polystyrene beads (refractive index $n = 1.6$, radius $r = 40$ nm) in (**d**) and back scatterers, i.e., silver nanoparticles in (**e**). Scale bar: 200 nm.

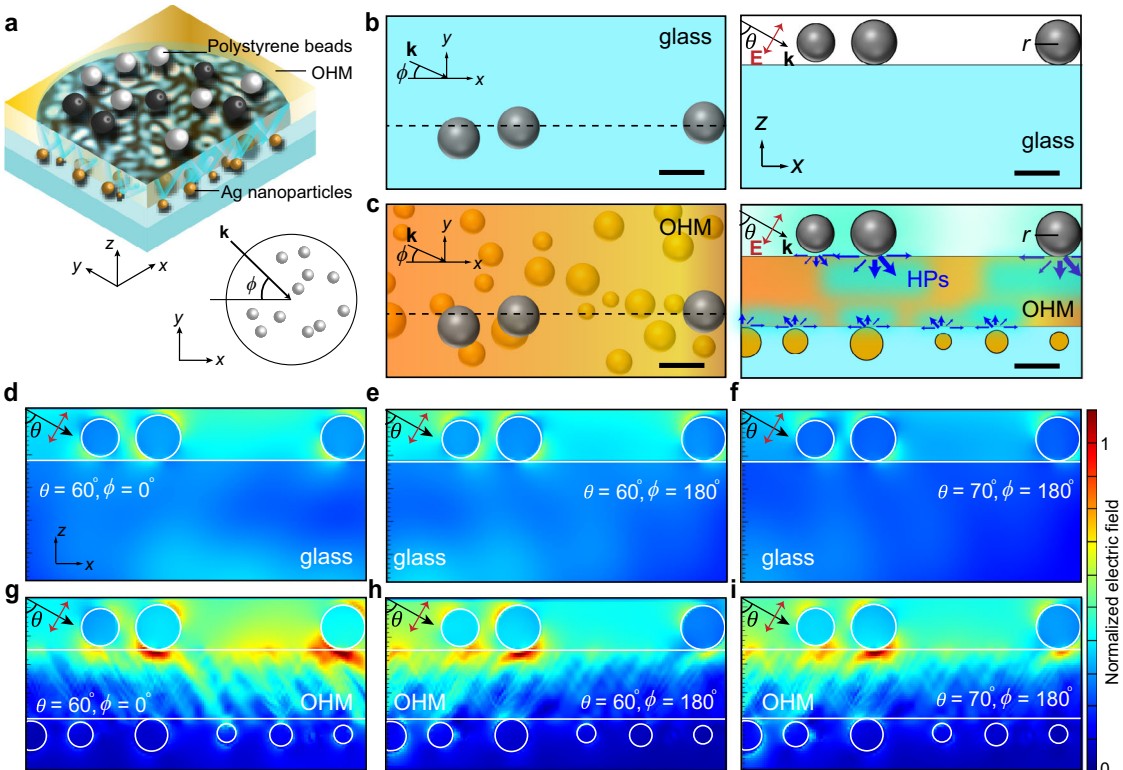

**Fig. 2 | Tailored scattering of dielectric nanoparticles by OHMs. a–c** Schematics of light scattering by dielectric nanoparticles (polystyrene, refractive index $n = 1.6$, radius $r = 40$ nm) situated on glass (**b**) and on OHM (**c**). Scale bar: 100 nm. **d–i** FDTD-calculated near-field intensity distribution in the vicinity of the dielectric nanoparticles on the glass (**d–f**) and on the OHM (**g–i**), where linearly (p-) polarized laser beam with electric field, **E** (red arrows), wave vector, **k** (black arrows), the polar angle, $\theta$, and azimuthal angle, $\phi$, was used as the excitation source at $\lambda = 532$ nm.

the propagation of low-loss hyperbolic polaritons (HPs)[36] with a sub-diffraction-limited light confinement in the visible range of 420–560 nm. The scattering of light by a scatterer on top of the OHM will be tailored due to the near-field coupling[41] between the scattering field and the HPs, as illustrated in Supplementary Fig. S1b. As a consequence, the far field scattering signal may get more than an order of magnitude enhancement compared to the case when the same scatterer sitting on a glass substrate, as shown in Supplementary Fig. S1c. This scattering enhancement is highly beneficial for improved signal-to-noise ratio in experiment, enabling high-contrast imaging for smaller objects.

The silver nanoparticle layer (Fig. 1e) randomly self-assembled at the bottom surface of the OHM (see Methods for details) helps to generate rich super-resolution speckles on the top surface mediated by high-$k$ HPs of the OHM when various incident angles are introduced (Fig. 2a–c). Compared to the case of glass samples (Fig. 2d–f), finite difference time domain (FDTD)-calculated near-field intensity distribution in the vicinity of the polystyrene beads (Fig. 2g–i) represents a tunable, strong, volumetrically confined near-field. This eventually leads to various far-field images of polystyrene beads with high scattering intensity (Supplementary Fig. S2). The speckle patterns resulting from the variations in the incident beam generate intensity variations in the high-contrast scattering images of the polystyrene beads (Supplementary Fig. S2d, e). The resultant high-contrast, partial scattering images of the polystyrene beads with various incident angles were recorded by a sCMOS camera with 80×/0.6 NA objective lens (see Methods for details).

Normally, the dielectric nanoparticles situated on top of a glass substrate experience multiple interferences under coherent illumination, which significantly reduces the contrast and makes them difficult to detect[13–16]. However, in the proposed HMES nanoscopy, the strongly

enhanced $E$-field in the vicinity of the polystyrene beads on the OHM reduces the effective contribution from interference fields away from the beads, leading to exceptionally high-contrast. Because the near-field illumination speckles comprise high-$k$ information which results from the excitation of HPs of the OHM, the centroid of the scattering intensity pattern by adjacent dielectric nanoparticles is shifted by a small perturbation in the illumination angle due to the difference in phase delay, indicating that the recorded partial scattering images can be used for super-resolution image reconstruction, as different dielectric nanoparticles provide distinctly different scattering profile (see Supplementary Fig. S2e).

## Image reconstruction for the HMES nanoscopy

The Blind-SIM reconstruction algorithm presented in ref. 39 was adopted for the image reconstruction in the HMES nanoscopy. To verify the suitability of using this algorithm for the HMES nanoscopy, partial scattering images of a ground truth sample were simulated by a 3D FDTD method (see Methods for details) and its super-resolution image was retrieved from these simulated partial scattering images by the Blind-SIM algorithm. Figure 3a shows the simulated partial scattering images of a given distribution of dielectric nanoparticles (Fig. 3b) situated on top of the OHM, illuminated from various directions ($\phi$ ranging from 0° to 330° with an interval of 30° and $\theta$ ranging from 60° to 69° with an interval of 1°). The incoherently averaged image from the $m \times n = 120$ partial scattering images are shown in Fig. 3c, where $m$ is the number of varied $\phi$ and $n$ is the number of varied $\theta$. The $m \times n$ partial scattering images were used for the Blind-SIM reconstruction. Figure 3d shows the retrieved image where adjacent particles with a center-to-center distance of 87 nm can be clearly resolved. Our simulation results in Supplementary Figs. S3–S6 confirm that the HMES method is possible to extend the resolution to the scale

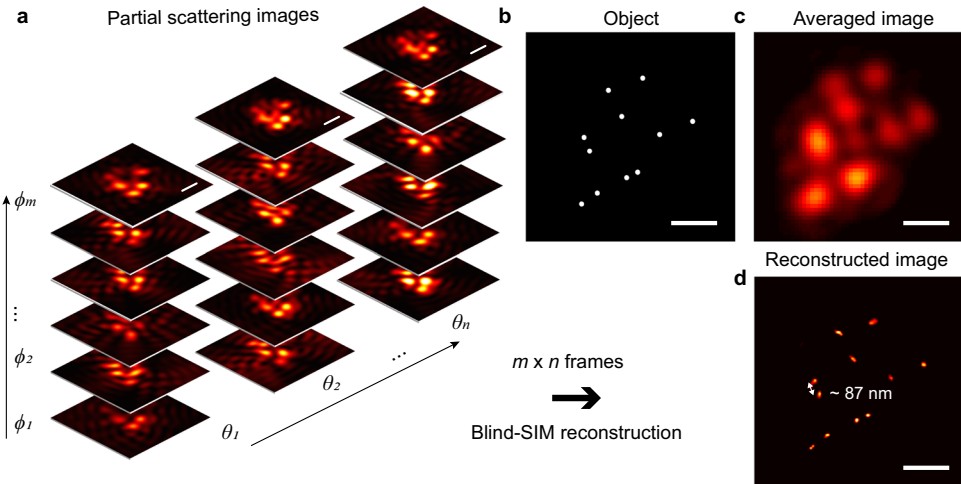

**Fig. 3 | Numerical simulation of the HMES nanoscopy. a** Partial scattering images of dielectric nanoparticles (radius $r = 40$ nm) with different oblique illumination angles, where the number of varied azimuthal angle of incidence $m = 12$, the number of varied polar angle of incidence $n = 10$. **b** The testing object. **c** A scattering image averaged over 120 images with coherent illumination from various directions. **d** Blind-SIM reconstructed image obtained from the 120 partial scattering images. Scale bars: 300 nm.

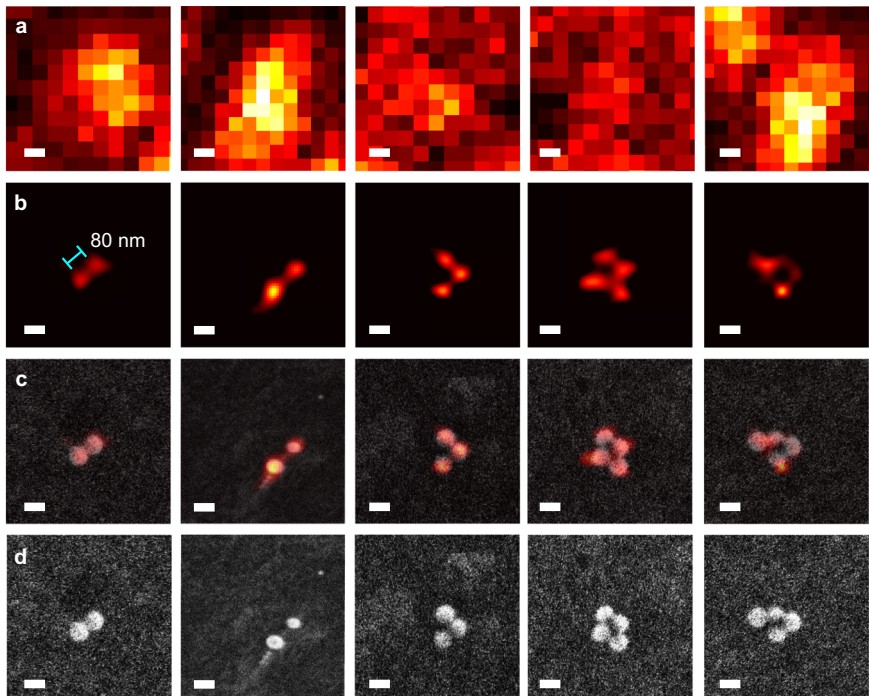

**Fig. 4 | HMES nanoscopy images of 80-nm polystyrene beads. a** Averaged partial scattering images. Total 240 partial scattering images were generated from 12 different azimuthal angular illuminations ($\phi$ ranging from 0° to 330° with an interval of 30°) with 20 varying speckle illumination patterns at $\theta = 60°$. **b** Super-resolution label-free scattering images of the polystyrene beads using the HMES nanoscopy. **c**, **d** Overlay of the HMES images and SEM images (**c**), and the SEM images (**d**). Scale bar: 100 nm.

of 40 nm by introducing smaller silver nanoparticles in the back scattering layer.

## HMES nanoscopy of polystyrene beads

As discussed above, with the HMES nanoscopy, partial scattering images with coherent illumination from various directions can be used to obtain the reconstructed image by the Blind-SIM algorithm. High-contrast, super-resolution label-free images are obtained from post-processing of the 240 frames. Figure 4a shows the experimentally obtained partial scattering images which are incoherently superposed with a diffraction-limited resolution. The HMES nanoscopy image is shown in Fig. 4b, where the adjacent 80 nm polystyrene beads are

clearly resolved. The corresponding SEM images are shown as ground truths in Fig. 4c (overlay of the HMES images and SEM images) and Fig. 4d (SEM images only). It is worth noting that the HMES nanoscopy can be used to distinguish the size of objects as shown in Supplementary Fig. S7.

We also investigated the image quality with respect to the number of frames $N$ used in our reconstruction and the results are shown in Fig. 5. The HMES nanoscopy dataset of the beads under 12 different azimuth angular illuminations with 60 small perturbation angles along the polar direction at $\theta = 60°$ is generated through a scanning mirror and a custom radial polarizer consisting of 12 polarizing stripes (see Fig. 5b and Methods for details). In the experiment, 720 partial

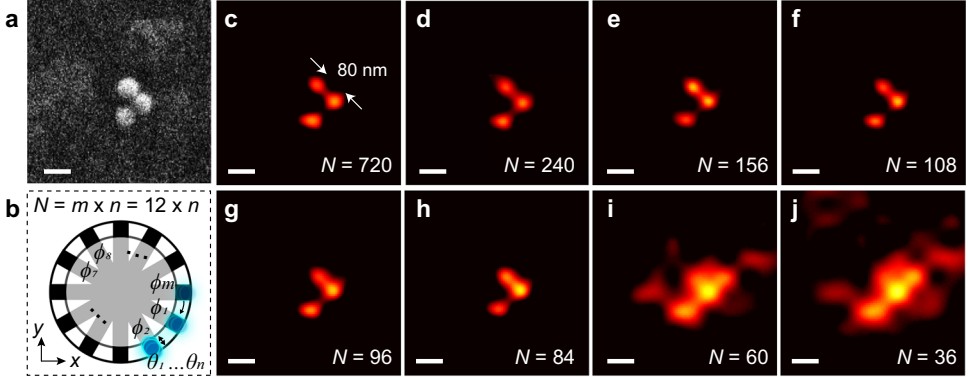

**Fig. 5 | Comparison of reconstructed images based on different numbers of frames in an experimental demonstration of HMES nanoscopy. a** SEM image of three polystyrene beads of 80 nm in diameter. Scale bar: 100 nm. **b** The BFP configuration of the HMES nanoscopy. Total $N$ partial scattering images were generated from $m = 12$ different azimuthal angular illuminations ($\phi$ ranging from 0° to 330° with an interval of 30°) with $n$ varying speckle illumination patterns (i.e., $n$ small perturbation angles along the polar direction) at $\theta = 60°$. **c–j** Reconstructed images based on different numbers of frames, **c** $N = 720$ ($n = 60$), **d** 240 ($n = 20$), **e** 156 ($n = 13$), **f** 108 ($n = 9$), **g** 96 ($n = 8$), **h** 84 ($n = 7$), **i** 60 ($n = 5$), and **j** 36 ($n = 3$). Scale bar: 100 nm.

scattering images are recorded at 10 Hz with an illumination intensity of approximately 7 W/cm². Clearly, the HMES nanoscopy needs to collect a sufficient number of frames to reconstruct the image with high resolution and fidelity, while the image quality will be saturated after an optimized number of frames which is somewhere between 100 and 200 (Fig. 5c–j). As one can imagine, the optimized $N$ is also dependent on the achievable resolution. At minimum exposure times of >2 ms, the label-free image with 80 nm resolution is guaranteed at $N > 80$ frames.

## Discussion

Enhancing the resolution of scattering microscopies is the main task of this work, so to have a better understanding of the resolution enhancement mechanism behind the proposed HMES nanoscopy, a theoretical framework adapted from refs. 42 and 43 is provided in Supplementary Section S8. According to the theoretical framework, the resolution enhancement of scattering imaging is $\frac{1}{2}\left(1 + \frac{k_{mat}}{NA \times k_0}\right)$, where $k_{mat}$ and $k_0$ are the maximum spatial frequencies supported by the material used and free space, respectively (Supplementary Table S1). Since high refractive-index materials support a large $k_{mat}$ and thus give rise to a high-resolution enhancement, they are the dominant factor of scattering image's resolution; unfortunately, as a result of lack of a high refractive-index material, current scattering imaging techniques have only demonstrated a resolution enhancement of ~3 fold[24–27] (Supplementary Table S2), as mentioned above. In the HMES nanoscopy demonstrated in this work, the high-$k$ low-loss hyperbolic material OHM (Supplementary Fig. S8) is used, so a 5.5-fold resolution enhancement is enabled in scattering microscopies. It is also worth pointing out that the resolution enhancement for both the high-$k$ hyperbolic metamaterials-based scattering or fluorescence nanoscopies could be further pushed with an increased detection NA, such as using a contact microsphere[44–47] or coupling whispering-gallery modes in the microsphere[48].

Artifacts in the final reconstructed image of HMES nanoscopy can be caused by several factors, such as directional scattering of densely packed objects, in addition to image reconstruction-related artifacts and lack of sufficient frames for reconstruction. A systematic study to deal with the artifacts is thus required in the future for obtaining a high-quality reconstructed super-resolution image. Such a study is expected to include identifying artifacts, figuring out their causes, and, if possible, objectively correcting for some common ones, in various application scenarios[49]. The theoretical framework provided in Supplementary Section S8 might be used as a guidance to clarify possible sources that cause the artifacts, and thus benefiting the future systematic study.

The focus of the HMES nanoscopy is on the surface imaging since the excited evanescent field decreases exponentially with the distance from the OHM. The scattering of small scatterers close to the OHM is selectively enhanced by the evanescent field illumination, and thus results in a high-contrast sectioned scattering image of the on-surface sample. Such a near-field scattering imaging can be very useful for observing structural details near the imaging object surface. However, for a small object—an aggregated cluster for example—that is away from the OHM surface, the effect of the high-$k$ component of the illumination pattern is less pronounced so that the scattering signal from this out-of-surface object becomes weak.

In summary, we have proposed and demonstrated HMES nanoscopy, a label-free super-resolution imaging technique. It is based on tailored scattering properties of dielectric nanoparticles situated on top of an OHM substrate which supports HPs. From the excitation of the HPs, field intensity in the contact area of the objects and the OHM substrate is strongly enhanced, providing a scattering signal that is clearly discernible. Partial scattering images with coherent illumination from various directions can be sequentially collected to apply the Blind-SIM algorithm. A deconvoluted scattering image by Blind-SIM reconstruction shows a center-to-center resolution of 80 nm. According to our numerical simulations, HMES nanoscopy promises a resolution of approximately 40 nm or even better with improved OHM properties. The demonstrated HMES nanoscopy is a potentially fast imaging method due to the high signal-to-noise ratio. HMES may lead to numerous potential applications in the field of label-free super-resolution imaging.

## Methods

### Sample fabrication

Ag nanoparticles (AgNPs) were synthesized via a polyol synthetic method[50] and purified with vacuum filtration (using 650, 450 and 220 nm pore size Millipore Durapore membranes). 1 mL AgNPs (stored in ethanol) were washed twice with ethanol and re-dispersed in $CHCl_3$ solution (~0.2 mL). A glass petri dish was filled with DI water, AgNPs (in $CHCl_3$) were added drop by drop to the air-water interface in petri dish and form a Langmuir film of AgNPs. After making the AgNPs film, we waited about 2 h till equilibrium and then transferred the AgNPs film onto an oxygen-plasma treated coverslip by dipping into the petri dish. Finally, a random array of AgNPs was formed on the coverslip. The average size of the AgNPs was 60 ± 30 nm. The variation in the shapes and sizes will provide richer dynamic speckles for HMES nanoscopy. Polymethylmethacrylate (950PMMA A2, Micro Chem) spin-coated thin film was formed with 2000 rpm for 45 s in order to reduce surface

roughness. After that, the rr-OHM film was fabricated[36]. 98% regioregular P3HT (molecular weight Mw ~87,000, Sigma Aldrich) of 100 mg was dissolved in 1 mL chlorobenzene by heating the solution to 50 °C for 3 h and resting at room temperature for 2 h. Thin films were spin-coated on the AgNPs-PMMA layer at 5000 rpm for 60 s. Optical characterization of the OHM (film thickness: 182 nm) was performed via a rotating polarizer type spectroscopic ellipsometer (J.A. Woollam M-2000D, J. A. Wollam Co. Ltd.)[36,51]. The optical transfer function calculation results show that the resolution improvement will be decreased with further increasing the thickness of the OHM film due to the absorption loss (Supplementary Fig. 8). Polystyrene beads (refractive index $n = 1.6$, radius $r = 40$ nm) were then drop-cast onto the OHM.

### Experiment setup

We used home-modified fluorescence microscopy (Olympus IX83) with an inverted configuration. A 532 nm laser (Ventus, MPC6000) was coupled into a multimode fiber (Thorlabs, core diameter: 50 μm, NA 0.2). The incidence angle of the beam was controlled by a pair of galvo mirrors (Thorlabs, GVS002). The $4f$ system projects the focused multimode beam on to the back focal plane of the objective lens, which allows for precise control of illumination angles (polar angle, $\theta$-60°, and varying azimuthal angle, $\phi$). A custom radial polarizer consisting of 12 polarizing stripes ensures $p$-polarization for different illumination angles[20] (Fig. 5b). To ensure high contrast and high efficiency, the $p$-polarization is required for near-field coupling between the scattering field and the high-$k$ HPs. Projecting the diffraction-limited, coherent speckle patterns on the specimens-OHM film will convert the pattern into high-resolution speckles that illuminate the specimens, where the input fiber spool is gently stretched by a step motor for each incidence azimuthal angle of illumination and imaging acquisition. A total of 60 varying dynamic speckle patterns were generated and illuminated specimens-OHM film by scanning an illumination angle $\phi$ ranging from 0° to 360°. For back focal plane filtering and dark-field detection, a diaphragm was mounted in an external Fourier plane (Hamamatsu Gemini 2C). A field lens unit and Bertrand lens unit were used to make sure the diaphragm was positioned properly. Reflected un-scattered light is almost perfectly blocked by the diaphragm in the pupil plane, only the scattered light generated from the specimens-OHM passes the diaphragm (iris) and forms the scattering image. A sCMOS camera was used for imaging acquisition (Hamamatsu ORCA Flash 4.0 v3) with an 80×/0.6 NA objective lens. To synchronize all equipment properly, we used MATLAB software to control a DAQ voltage output module (NI-9263) from National Instruments.

### FDTD simulation

A 3D FDTD simulation was performed using a commercial software package (FDTD, Lumerical). Perfectly matched layer (PML) boundary conditions were set up in the $x$-, $y$- and $z$-direction. The permittivity of polystyrene and experimentally obtained permittivity of 180 nm thick OHM were employed for the calculations. To calculate the scattering enhancement of a polystyrene nanoparticle on the OHM compared to that on the glass, a total-field/scattered-field (TFSF) source with $p$-polarization was incident from the top side of the sample. By monitoring the outgoing scattered field intensity from the polystyrene nanoparticle, scattering cross-section was calculated on OHM and that on glass, separately. To calculate near-field intensity distributions in the vicinity of the polystyrene nanoparticles on the OHM, a 2D $xz$-plane power monitor was used. The total field and scattered field were monitored by three 2D $xy$-plane power monitors—0, 2, 20 nm above the polystyrene nanoparticles on OHM. Far-field scattering images of the polystyrene nanoparticles on OHM were calculated from the data of near field

monitors to obtain the microscopic images. After the far-field projection of the intensity (i.e., decomposition of near-field into plane waves)[52,53], any plane waves with angles outside of the NA = 1 are then discarded, and the light is re-focused onto an image plane, using chirped $z$-transform[54] (a generalization of the discrete Fourier transform). These results have been post-processed by the blind-SIM algorithm with MATLAB.

### Imaging reconstruction

All our image processing and reconstruction are performed in MATLAB. An iterative reconstruction algorithm, blind-SIM[39] which does not require exact knowledge of the illumination pattern, is used to retrieve the object information. For blind-SIM, an assumption is made that all illumination patterns add up to a uniform pattern. Both the object and the illumination patterns are treated as unknowns in real space and are solved using a cost-minimization strategy. The GPU based reconstruction typically takes 10 min on a Nvidia GTX 1080 Ti.

## Data availability

The raw image files used in this paper are available on https://www.zliugroup.com/data/Public/Blind_SIM_packages.html. The data that support the findings of this study are available on request from the corresponding author.

## Code availability

The custom-written MATLAB code for speckle-MAIN reconstruction is available on https://www.zliugroup.com/data/Public/Blind_SIM_packages.html.

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

## Acknowledgements

This work was supported by the Gordon and Betty Moore Foundation (Z.L.). Y.U.L. acknowledges support by "Regional Innovation Strategy (RIS)" through the National Research Foundation of Korea (NRF) funded by the Ministry of Education (MOE) (2021RIS-001) and NRF-2014M3A6B3063708. G.B.M.W. acknowledges 2 years scholarship from LPDP (Indonesia Endowment Fund for Education) for pursuing master degree in United States. A.R.T. acknowledges support from the National Science Foundation (CHE1807891).

## Author contributions

Y.U.L. and Z.L. conceived this study. Y.U.L., S.L., G.B.M.W., J.Z., and Y.Z. performed the experiments. G.B.M.W., J.Z., and Y.Z. prepared samples under the supervision of A.R.T and Z.L. Y.U.L. performed the simulations and reconstructed the images. All authors contributed to scientific discussions and analyzed the data. Y.U.L. created the figures. Y.U.L. and S.L. wrote the paper which was revised by all authors.

## Competing interests

The authors declare no competing interests.
