## [Peer Review File · Nature Communications]

Hyperbolic material enhanced scattering nanoscopy for label-free super-resolution imagingReviewers' comments:

Reviewer #1 (Remarks to the Author):

In this article, the authors have demonstrated a new method for far-field label-free super-resolution imaging. The key point of this method is the use of organic hyperbolic material as the substrate for the specimens. The new substrate not only enhances the scattering intensities of small objects but also controls the illumination/scattered field distributions at deep sub-wavelength scales. Combined with the blind structured illumination microscopy reconstruction algorithm, the proposed method can resolve adjacent nanoparticles with a center-to-center resolution of 80 nm. In my opinions, the work presented in this manuscript is very interesting and useful, the experimental and theoretical results are solid and repeatable, so I recommend publication in the journal Nature Communications after that the authors address the following concerns.

1. In Figure 1a, a radial polarizer was used, and in Figure 5b show a custom polarizer consisting of 12 polarizing stripes. Are they the of same element? Can the authors present more information on this polarizer? Is it a home-built component or a commercial product? Can this radial polarizer be replaced with a linear-polarizer and a quarter waveplate, so that the incident beam is of circular polarization? It seems that the circular polarized incident beam is also suitable for this experiment.

2. In this experiment, 60 ± 30 nm silver nanoparticles were randomly self-assembled at the bottom surface of the substrate to provide more various high-k illuminations via small perturbations in the illumination. Is the size of the silver nanoparticles related with the imaging resolution or the size of the nanoparticles to be resolved? Can the smaller silver nanoparticles will introduce higher-k illumination and then result in better imaging resolution?

3. I understand that the use of high-k illuminations enabled by the OHM and silver nanoparticle will result in imaging of very smaller features of the specimens, but it will probably induce the loss of information of the larger features of the specimens. For example, as shown in Figure S3d, the shape of the imaged bend wires is not precisely identical to the objects. Can the authors give some descriptions on this point?

4. Can the proposed method be used to distinguish the size of the nanoparticles? In the experiments, the imaged beads are of the same diameters, and what will happen if the beads of different size? Can this method be used to resolve the close distance between two beads of different diameters?

5. I wonder if the illumination from the bottom surface of the substrate will provide a better imaging contrast.

6. It seems that if the deep learning method was used, a smaller number of the images was needed for the reconstructions, and then the imaging speed will be higher. Is it right?

Reviewer #2 (Remarks to the Author):

Yeon Lee et.al in this manuscript report a method to enhance the resolution of label-free

optical microscopy. The authors claims that coherent illumination can be averaged at the camera plane to produce speckle-free diffraction limited images. Moreover, by exploiting the high spatial frequencies generated by hyperbolic materials together with Blind-SIM reconstruction algorithm the resolution of the label-free method can be further pushed beyond the diffraction limit.

The paper is built on recent progress towards the rapid growing field of label-free super-resolution optical microscopy. The main concern, I have towards this paper is lack of novelty both from the conceptional level and also on the achievable results. High contrast and super-resolution label-free microscopy have now been demonstrated by several groups. Also, contrary to what authors claim in their paper “«current label-free scattering imaging techniques have only demonstrated resolution around 150 nm», several groups have reported sub-100 nm results in label-free mode.

Firstly, the fundamental idea to average several coherent scattering images to generate high-contrast and high-resolution images are well understood and covered in the literature starting from ROCs and several other papers thereafter (some of them mentioned below). All these works are based on near-field excitation similar to what authors performed in this manuscript. Also, it is well known in the optics community that using hyperbolic materials, plasmons, or high-refractive index substrate, it is possible to generate high-spatial near-field frequencies that can be exploited for the super-resolution application. Finally, to obtain super-resolution within the label-free context, several groups have demonstrated that it is possible to exploits different super-resolution algorithms such as Fourier ptychography, or other intensity fluctuation to obtain sub-100 nm resolution.

Thus, in the present work, authors have demonstrated another approach via different substrate coupled with Bind-SIM approach to proof what is already been demonstrated by several other groups, with similar results. My understanding of high-impact journal such as Nature Communication is that the work should be based on novel idea, I leave this to the Editor to decide what level of innovation is adequate to fulfil this criterion. My impression is that this paper is more suitable for Scientific Reports that demands systematic simulation and experimental results and is less rigorous on novelty.

Some papers that have demonstrated super-resolution label-free nanoscopy:

1. Mingwei Tang, et.al., “Scalable universal tunable virtual-wavevector spatial frequency shift (TVSFS) super-resolution imaging» arXiv:2103.09321

In this work, resolution enhancement of 5X is reported in label-free context. Two beads resolved with 93 nm apart.

2.X.W.Liu,C.F.Kuang,X.Hao,C.L.Pang,P.F.Xu,H.F.Li,Y.Liu,C.Yu,Y.K.Xu,D.Nan,W. D. Shen, Y. Fang, L. N. He, X. Liu, Q. Yang, Phys. Rev. Lett. 2017, 118, 076101.

Here, structured down to 70 nm are imaged with 0.85 N.A. objective lens. The achievable resolution is again very similar results to what is reported in this work.

3. C. Pang, J. Li, M. Tang, J. Wang, I. Mela, F. Ströhl, L. Hecker, W. Shen, Q. Liu, X. Liu, Y. Wang, H. Zhang, M. Xu, X. Zhang, X. Liu, Q. Yang, C. F. Kaminski, On-Chip Super-Resolution Imaging with Fluorescent Polymer Films, Adv. Funct. Mater. 2019, 29, 1900126.

4. F. Ströhl, I. S. Opstad, J.-C. Tinguely, F. T. Dullo, I. Mela, J. W. M. Osterrieth, B. S.

Ahluwalia, C. F. Kaminski, Opt. Exp. 2019, 27, 25280.

Above mentioned papers have used different approaches to achieve super-resolution imaging in label-free domain, several below 100 nm as opposed to what authors claimed in the paper.

Further, I recommend following revision that could assist in polishing the content of the manuscript:

a) The impact of this work is mostly in biology; however, this work focuses primarily on nanobeads made of polystyrene, that have relatively high-refractive index. Authors are encouraged to perform bio-imaging experiments, where scattering will be limited due to low refractive index contrast of sub-cellular organelles in media and cytosol. Here, the main concern is on using object/sample with small refractive index contrast.

b) It will be beneficial for the readers to have a discussion on issues which could arise when imaging real 3D bio-samples. The concept presented is near-field effect, I foresee there will be challenges and limitation when a real sample, e.g. cells are imaged. This is mainly due to loss of resolution due to "multiple-scattering" from different organelles, which will limit the true reconstruction in the far-field.

c) Further, it will be interesting to discuss or simulate how does Blind-SIM reconstruction will work when the scattering is sufferings from "multiple-scattering" from objects that are not within the exponential decaying near-field component.

d) Both the points b and c, would be even more critical in the present upright microscopy configuration, where the light has to travel from the rest of the sample before it can be imaged by the objective lens.

e) Is there any limitation of not using oil-immersion objective lens? Possible due to use of upright microscopy?

f) It is advised to make the raw files and Blind SIM algorithm available as open access for readers. The main known challenge with Blind SIM is the reconstruction artefacts and making the raw files and the algorithm available would make this work more accessible and enable wider penetration.

g) It will also be useful for readers to have more information about OHM. Can it be re-used? What is the optimal thickness of this layer? How and if the resolution obtained is dependent on the thickness of the layer.

We thank both reviewers for their positive evaluation of our work and for their valuable comments on this manuscript. A one-to-one response to these comments with the corresponding manuscript modifications are given below, which are rendered in a different text format for clarity:

Blue Italic: Reviewer's comment;
Black: Our response;
Red: Modifications of the manuscript.

Response to Reviewer #1

In this article, the authors have demonstrated a new method for far-field label-free super-resolution imaging. The key point of this method is the use of organic hyperbolic material as the substrate for the specimens. The new substrate not only enhances the scattering intensities of small objects but also controls the illumination/scattered field distributions at deep sub-wavelength scales. Combined with the blind structured illumination microscopy reconstruction algorithm, the proposed method can resolve adjacent nanoparticles with a center-to-center resolution of 80 nm. In my opinions, the work presented in this manuscript is very interesting and useful, the experimental and theoretical results are solid and repeatable, so I recommend publication in the journal Nature Communications after that the authors address the following concerns.

[Reply] We appreciate the reviewer's concise summary of our work and his/her recommendation for publication.

[Question 1-1] *In Figure 1a, a radial polarizer was used, and in Figure 5b show a custom polarizer consisting of 12 polarizing stripes. Are they the of same element? Can the authors present more information on this polarizer? Is it a home-built component or a commercial product? Can this radial polarizer be replaced with a linear-polarizer and a quarter waveplate, so that the incident beam is of circular polarization? It seems that the circular polarized incident beam is also suitable for this experiment.*

[Reply 1-1] We thank the reviewer for the valuable comments. Yes, the radial polarizers mentioned in Figure 1a and Figure 5b are the same one, which is a custom radial polarizer as shown in Figure R1. It was used to generate p -polarized light for various angles of incidence of the excitation beam within our $4f$ system. 12 strips of wire grid polarizing films were cut and placed in an axially symmetric configuration. The rotational symmetry of radially polarized light (see the red arrows in Figure R1) effectively provides a p -polarization with respect to the substrate. The p -polarization is required for the excitation of the high- k optical mode that has a strong field intensity within a small region between the OHM film and dielectric beads.

Figure R1. Custom radial polarizer.

In response to the comments, we have provided the following revisions in the revised manuscript:

[Revision in the main text]

On page 3, “The setup is based on a common reflection microscope with slight modifications. A high-speed scanning mirror system, a custom radial polarizer (see **Methods for details**), and a focusing lens are introduced such that the incident azimuthal angle ϕ of a p -polarized laser beam ($\lambda = 532$ nm) can be swept from 0° to 360° . Meanwhile, the polar angle, θ , is confined between 60° to 70° by focusing the laser beam on the edge of the back focal plane (BFP) of the objective lens.”

On page 7, “The HMES nanoscopy dataset of the beads under 12 different azimuth angular illuminations with 60 small perturbation angles along the polar direction at $\theta = 60^\circ$ is generated through a scanning mirror and a custom **radial** polarizer consisting of 12 polarizing stripes (see Figure 5b and **Methods for details**).”

On page 9 Methods section. “A custom radial polarizer **consisting of 12 polarizing stripes** ensures p -polarization at different illumination angles³² (Figure 5b) . **To ensure high contrast and high efficiency, the p -polarization is required for near-field coupling between the scattering field and the high- k HPs.**”

[Question 1-2] *In this experiment, 60 ± 30 nm silver nanoparticles were randomly self-assembled at the bottom surface of the substrate to provide more various high- k illuminations via small perturbations in the illumination. Is the size of the silver nanoparticles related with the imaging resolution or the size of the nanoparticles to be resolved? Can the smaller silver nanoparticles will introduce higher- k illumination and then result in better imaging resolution?*

[Reply 1-2] We thank the reviewer for pointing out the nanoparticle’s size-dependence of the imaging resolution. We have checked and confirmed that the use of smaller silver nanoparticles gives rise to speckle illumination patterns with a higher k and thus results in a better imaging resolution. We have provided the following revisions in the revised manuscript and Supporting Information:

[Revision in the main text]

On page 5, “Our simulation results in **Figure S5 and Figure S6** confirm that the HMES method

is possible to extend the resolution to the scale of 40 nm by introducing smaller silver nanoparticles in the back scattering layer.”

[Revision in Supporting Information]

S5. The role of silver nanoparticle size on the backside of the OHM

Figure S5. The role of silver nanoparticle size on the backside of the OHM. (a, b) High- k near-field illumination speckles on the sample plane generated by the OHM. The silver nanoparticles with diameters of (a) 80 nm and (b) 20 nm on the backside of the OHM were introduced. (c) Cross-section curves.

S6. Numerical simulation of HMES nanoscopy (3)

The HMES nanoscopy images of adjacent 40-nm polystyrene beads separated from 40 to 80 nm (Figure S6a) were simulated to verify the resolving capability of the HMES nanoscopy. The silver nanoparticles with a diameter of 20 nm on the backside of the OHM were introduced. The corresponding diffraction-limited scattering image is shown in Figure S6b. The reconstructed super-resolution scattering image based on the HMES nanoscopy is shown in Figure S6c. Two pairs (Figure S6d) and three pairs (Figure S6e) of closely located polystyrene beads are resolved with a distance ~ 40 nm.

[Question 1-3] *I understand that the use of high- k illuminations enabled by the OHM and silver nanoparticle will result in imaging of very smaller features of the specimens, but it will probably induce the loss of information of the larger features of the specimens. For example, as shown in Figure S3d, the shape of the imaged bend wires is not precisely identical to the objects. Can the authors give some descriptions on this point?*

[Reply 1-3] We thank the reviewer for the valuable comments. As the reviewer pointed out, the performance of a structured illumination-based microscopy can be limited by the missing information in the spatial frequency domain when the illumination patterns contain discrete spatial frequencies. We believe that randomly generated speckle illumination patterns—i.e. the blind-structured illumination—under dynamically changed angles of incidence can have a variety of spatial frequencies, including the low spatial frequencies concerned by the reviewer (please see the calculated OTF shown in Figure R2 below) when enough frames are used in our reconstruction algorithm (please see Figure 5). As can be seen, the OTF of the OHM has a broad spatial frequency range with a large cut-off spatial-frequency $k_{\text{cut-off}} = 13k_0$ when $\text{OTF}(k-$

cut-off) = 10^{-4} at $\lambda = 532$ nm, which indicates that the resolution supported by this OHM can reach below 40 nm. Notice the OTF of the very small k -vectors ($<1.5k_0$) of the OHM is relatively low, but those information can be collected by traditional diffraction limited optics.

In response to the comments, we have provided the following revisions in the revised manuscript:

[Revision in the main text]

On page 8, “Artifacts in the final reconstructed image of HMES nanoscopy can be caused by several factors, such as directional scattering of densely packed objects, in addition to image reconstruction-related artifacts and lack of sufficient frames for reconstruction.”

Figure R2. Calculated optical transfer function (OTF) in spatial frequency domain of a 180-nm thick OHM film.

[Question 1-4] *Can the proposed method be used to distinguish the size of the nanoparticles? In the experiments, the imaged beads are of the same diameters, and what will happen if the beads of different size? Can this method be used to resolve the close distance between two beads of different diameters?*

[Reply 1-4] We thank the reviewer for the valuable comments. Yes, the proposed method in this manuscript can be used to distinguish the size of the nanoparticles, as shown in the revised Figure S7 below. We have provided the following revisions in the revised manuscript and Supporting Information:

[Revision in the main text]

On page 7, “The HMES nanoscopy image is shown in Figure 4b, where the adjacent 80 nm polystyrene beads are clearly resolved. The corresponding SEM images are shown as ground truths in Figure 4c (overlay of the HMES images and SEM images) and 4d (SEM images only). It is worth noting that the HMES nanoscopy can be used to distinguish the size of objects as shown in Figure S7.”

[Revision in Supporting Information]

S7. HMES nanoscopy images of polystyrene beads with different diameters

Figure S7. HMES nanoscopy images of polystyrene beads with different diameters. (a) SEM image of two beads with diameters of $D1 = 72$ nm and $D2 = 52$ nm, respectively. (b) Super-resolution label-free scattering images of the polystyrene beads using the HMES nanoscopy. Scale bar: 100 nm. (c) Normalized intensity profiles along yellow paths in the SEM image (black curve) and the HMES image (red curve), respectively.

[Question 1-5] *I wonder if the illumination from the bottom surface of the substrate will provide a better imaging contrast.*

[Reply 1-5] We thank the reviewer for the valuable comment. We believe that the illumination from the bottom surface and detection from the top surface indeed may provide a better imaging contrast. Currently, our imaging system only allows the backscattering detection mode (i.e. either top illumination and top detection, or bottom illumination and bottom detection) with various incidence angles. We choose top illumination and top detection (Figure 1a), not bottom illumination and bottom detection, because in the latter case the scattering image will be disturbed by the gold particles in the back scattering layer.

[Question 1-6] *It seems that if the deep learning method was used, a smaller number of the images was needed for the reconstructions, and then the imaging speed will be higher. Is it right?*

[Reply 1-6] We thank the reviewer for the valuable comments. Yes, we believe that deep learning methods will be of great help in improving both detection speed and reconstruction speed by decreasing the necessary number frame for reconstruction, as the reviewer pointed out. That will be something we would like to implement in the near future.

Response for Reviewer #2

Yeon Lee et.al in this manuscript report a method to enhance the resolution of label-free optical microscopy. The authors claims that coherent illumination can be averaged at the camera plane to produce speckle-free diffraction limited images. Moreover, by exploiting the high spatial frequencies generated by hyperbolic materials together with Blind-SIM reconstruction algorithm the resolution of the label-free method can be further pushed beyond the diffraction limit.

The paper is built on recent progress towards the rapid growing field of label-free super-resolution optical microscopy. The main concern, I have towards this paper is lack of novelty both from the conceptional level and also on the achievable results. High contrast and super-resolution label-free microscopy have now been demonstrated by several groups. Also, contrary to what authors claim in their paper “«current label-free scattering imaging techniques have only demonstrated resolution around 150 nm», several groups have reported sub-100 nm results in label-free mode. Firstly, the fundamental idea to average several coherent scattering images to generate high-contrast and high-resolution images are well understood and covered in the literature starting from ROCs and several other papers thereafter (some of them mentioned below). All these works are based on near-field excitation similar to what authors performed in this manuscript. Also, it is well known in the optics community that using hyperbolic materials, plasmons, or high-refractive index substrate, it is possible to generate high-spatial near-field frequencies that can be exploited for the super-resolution application. Finally, to obtain super-resolution within the label-free context, several groups have demonstrated that it is possible to exploits different super-resolution algorithms such as Fourier ptychography, or other intensity fluctuation to obtain sub-100 nm resolution. Thus, in the present work, authors have demonstrated another approach via different substrate coupled with Bind-SIM approach to proof what is already been demonstrated by several other groups, with similar results. My understanding of high-impact journal such as Nature Communication is that the work should be based on novel idea, I leave this to the Editor to decide what level of innovation is adequate to fulfil this criterion. My impression is that this paper is more suitable for Scientific Reports that demands systematic simulation and experimental results and is less rigorous on novelty. Some papers that have demonstrated super-resolution label-free nanoscopy:

1. Mingwei Tang, et.al., “Scalable universal tunable virtual-wavevector spatial frequency shift (TVSFS) super-resolution imaging» arXiv:2103.09321

In this work, resolution enhancement of 5X is reported in label-free context. Two beads resolved with 93 nm apart.

2. X. W. Liu, C. F. Kuang, X. Hao, C. L. Pang, P. F. Xu, H. F. Li, Y. Liu, C. Yu, Y. K. Xu, D. Nan, W. D. Shen, Y. Fang, L. N. He, X. Liu, Q. Yang, Phys. Rev. Lett. 2017, 118, 076101.

Here, structured down to 70 nm are imaged with 0.85 N.A. objective lens. The achievable resolution is again very similar results to what is reported in this work.

3. C. Pang, J. Li, M. Tang, J. Wang, I. Mela, F. Ströhl, L. Hecker, W. Shen, Q. Liu, X. Liu, Y. Wang, H. Zhang, M. Xu, X. Zhang, X. Liu, Q. Yang, C. F. Kaminski, On-Chip Super-Resolution Imaging with Fluorescent Polymer Films, Adv. Funct. Mater. 2019, 29, 1900126.

4. F. Ströhl, I. S. Opstad, J.-C. Tinguely, F. T. Dullo, I. Mela, J. W. M. Osterrieth, B. S. Ahluwalia, C. F. Kaminski, *Opt. Exp.* 2019, 27, 25280.

Above mentioned papers have used different approaches to achieve super-resolution imaging in label-free domain, several below 100 nm as opposed to what authors claimed in the paper.

[Reply] Discussion about novelty

We thank the reviewer for the comments. We have checked the references that the reviewer provided. By doing so, we found that the resolution of a superresolution method can easily be misjudged if only its absolute value is mentioned, as shown in the table below. In addition to point out the specific experimental conditions, such as the objective NA and the wavelength of the scattered light, we think the resolution improvement is a better way to assess the resolution of a superresolution method. According to the references that the reviewer mentioned, the state-of-the-art label-free scattering imaging techniques demonstrate only a 3-fold resolution improvement. We want to emphasize that this resolution improvement factor is fundamentally limited by the maximum k -vector supported by the material. Ref. 1 obtained 3-fold improvement simply because of the high index GaP can provide. In fact, GaP has been used for ~100 nm imaging in 2011 because of its high index (PRL 106, 193905, 2011). It is extremely hard to find any transparent natural materials with index larger than GaP's. In contrast, we experimentally demonstrate a label-free superresolution method with a 5.5-fold resolution improvement by using an organic hyperbolic material thin film made by simply spin-coating. **The novelty of our work relies on the fact that this is the first time to introduce OHM with optical properties much better than traditional optical materials. We applied it to the label-free super-resolution imaging and demonstrated the record high resolution improvement. In addition, as we illustrated above, the effective refractive index of OHM goes beyond 10, indicating the much larger image resolution improvement is possible.**

Ref.	Submission/ Publication date	Optical media	Lens NA, Wavelength, Diffraction-limited resolution	Absolute resolution	Resolution improvement	Used algorithm
Our work	11 Feb 2021	Organic hyperbolic materials	0.6 NA, $\lambda = 532$ nm, 443 nm	80 nm	~5.5 fold	Blind-SIM algorithm
1	arXiv 15 Mar 2021	Scalable photonic chip ($n = 3.43$) with grating	1.1 NA, $\lambda = 660$ nm, 300 nm	93 nm	~3 fold	Iterative algorithm
2	17 Feb 2017	Fluorescent nanowire ring	0.85 NA, $\lambda = 520$ nm, 306 nm	140 nm	~2 fold	Iterative algorithm
3	4 April 2019	Fluorescent polymer films	0.85 NA, $\lambda = 532$ nm, 313 nm	149 nm	~2 fold	Iterative algorithm
4	2 Sep 2019	Si ₃ N ₄ waveguide	0.95 NA, $\lambda = 488, 561, 647$ nm, 257, 295, 341 nm	160 nm	<2 fold	Fourier ptychography phase retrieval algorithm

We have thus revised the statement **on page 2**, from “current label-free scattering imaging

techniques have only demonstrated resolution around 150 nm.” to “**current label-free scattering imaging techniques have only demonstrated a resolution improvement of ~3 fold.**”, to emphasize the significant resolution improvement in our work.

We also noticed that, most of the references mentioned by the reviewer are from one research group using evanescent waves generated by dielectric waveguides. **In this manuscript, our focus is on the exploration of using hyperbolic materials**—which have become a rapidly growing research field and have been widely used for applications relying on light manipulation at sub-diffraction limited length scales, e.g. biosensor, super-resolution imaging, and broadband enhancement of spontaneous emission—**for label-free super-resolution imaging: This is also the main contribution of this manuscript.**

Moreover, we would like to emphasize that, a most recently published paper from the same group (“Structured illumination microscopy using a photonic chip” (Mingwei Tang, et.al., arXiv:2103.09321) reports ~3 times enhancement in imaging spatial resolution by using near-field excitation in a photonic waveguide. The photonic waveguide can generate higher resolution illumination patterns compared to patterns generated in free space purely because of the high refractive index of the “dielectric materials” composing the photonic waveguide (refractive index of the waveguide, $n = 3.43$). Therefore, it is necessary to introduce plasmonic materials/metamaterials for better resolution improvement. To the best of our knowledge, our study is the first experimental demonstration of super-resolution label-free imaging with high- k ($\sim 13k_0$, where k_0 is the wavevector of incident beam) speckle illumination generated by metamaterials.

We thank the reviewer for sharing his/her understanding of high-impact journal. We agree with the reviewer that we should “*leave this to the Editor to decide what level of innovation is adequate to fulfil this criterion*”. **Here, we would like to re-emphasize the novelty of this manuscript: We experimentally demonstrate a label-free superresolution method with a 5.5-fold resolution improvement beyond the diffraction limit by using an organic hyperbolic material thin film made by simply spin-coating. The resolution improvement factor has great potential to go beyond 10 according our estimation. We believe that such a simple but efficient label-free superresolution approach will gain a lot of attention from the optical microscope community, and thus bring a high impact on the relevant research fields.**

[Question 2-1] *Further, I recommend following revision that could assist in polishing the content of the manuscript: a) The impact of this work is mostly in biology; however, this work focuses primarily on nanobeads made of polystyrene, that have relatively high-refractive index. Authors are encouraged to perform bio-imaging experiments, where scattering will be limited due to low refractive index contrast of sub-cellular organelles in media and cytosol. Here, the main concern is on using object/sample with small refractive index contrast.*

[Reply 2-1] We thank the reviewer for the encouragement of performing bio-imaging experiments. **The novelty of this manuscript is the first experimental demonstration of label-free superresolution imaging using organic hyperbolic material thin films with a**

5.5-fold resolution improvement beyond the diffraction limit. Bio-imaging experiments using the approach demonstrated here are surely worth to do, which, however, out of the scope of this manuscript. **As mentioned above, we believe that such a simple but efficient label-free superresolution approach will gain a lot of attention from the optical microscope community, such that the bio-imaging experts will follow a series of systematic studies on the bio-experiments — by this way, the high impact of this manuscript is spread to the relevant research fields.**

It is a big challenge for label-free imaging of small objects with a subwavelength size due to the wave nature of light, which is well described by the Rayleigh scattering: The scattering cross-section of a Rayleigh scatterer is proportional to the inverse fourth power of the wavelength and the sixth power of its size. This challenge is even more pronounced when “object/sample with small refractive index contrast”, as concerned by the reviewer. As the consequence, there are no practical solutions for all the previous works (such as the references mentioned by the reviewer) when the objects are getting smaller and less contrast to the environment. This is the reason why “current label-free scattering imaging techniques have only demonstrated a resolution improvement of ~3 fold”. However, our work using hyperbolic materials demonstrates exceptionally enhanced scattering signals and thus a label-free superresolution imaging with a 5.5-fold resolution improvement. **Considering the inverse fourth power dependence of the scattering cross-section on the wavelength, the 5.5-fold resolution improvement by our work is remarkable, which thus opens a practical way to address the big challenge of label-free imaging of a tiny object with a small refractive index contrast.**

[Question 2-2] b) It will be beneficial for the readers to have a discussion on issues which could arise when imaging real 3D bio-samples. The concept presented is near-field effect, I foresee there will be challenges and limitation when a real sample, e.g. cells are imaged. This is mainly due to loss of resolution due to “multiple-scattering” from different organelles, which will limit the true reconstruction in the far-field. c) Further, it will be interesting to discuss or simulate how does Blind-SIM reconstruction will work when the scattering is sufferings from “multiple-scattering” from objects that are not within the exponential decaying near-field component.

[Reply 2-2] Discussion about 3D imaging sufferings from multiple scattering from objects.

We thank the reviewer for the comment. We agree with the reviewer that it is worth discussing the near-field effect of super-resolution speckles in this HMES nanoscopy work. We have provided the following revisions in the revised manuscript:

[Revision in the main text]

On page 8, “The focus of the HMES nanoscopy is on the surface imaging since the excited evanescent field decreases exponentially with the distance from the OHM. The scattering of small scatterers close to the OHM is selectively enhanced by the evanescent field illumination, and thus results in a high-contrast sectioned scattering image of the on-surface sample. Such a

near-field scattering imaging can be very useful for observing structural details near the imaging object surface. However, for a small object—an aggregated cluster for example—that is away from the OHM surface, the effect of the high- k component of the illumination pattern is less pronounced so that the scattering signal from this out-of-surface object becomes weak.”

[Question 2-3] *d) Both the points b and c, would be even more critical in the present upright microscopy configuration, where the light has to travel from the rest of the sample before it can be imaged by the objective lens.*

[Reply 2-3] We thank the reviewer for the comment. Currently, our imaging system only allows the backscattering detection mode (i.e. either top illumination and top detection, or bottom illumination and bottom detection) with various incidence angles. We choose top illumination and top detection (Figure 1a), not bottom illumination and bottom detection, because in the latter case the scattering image will be disturbed by the gold particles in the back scattering layer.

[Question 2-4] *e) Is there any limitation of not using oil-immersion objective lens? Possible due to use of upright microscopy?*

[Reply 2-4] We thank the reviewer for the questions. There is no fundamental limitation of not using an oil-immersion objective lens. In this proof-of-concept study, we had to obtain SEM ground truth images after the HMES nanoscopy imaging. Thus, the things that cause contamination of the sample, such as oil, water, and additional coverslips attached to the specially distributed polystyrene beads, were avoided.

[Question 2-5] *f) It is advised to make the raw files and Blind SIM algorithm available as open access for readers. The main known challenge with Blind SIM is the reconstruction artefacts and making the raw files and the algorithm available would make this work more accessible and enable wider penetration.*

[Reply 2-5] We thank the reviewer for the suggestion. We have added the sections of ‘Code availability’ and ‘Data availability’ in the revised manuscript.

[Question 2-6] *g) It will also be useful for readers to have more information about OHM. Can it be re-used? What is the optimal thickness of this layer? How and if the resolution obtained is dependent on the thickness of the layer.*

[Reply 2-6] We thank the reviewer for the suggestion. The OHM films are waterproof and washable with a good biocompatibility (please see our recent publications: *Adv. Mater.* 2021, 33, 2006496, *Adv. Mater.* 2020, 32, 2002387), which are thus reusable. We have provided the following revisions on the thickness of the OHM films in the revised manuscript and Supporting Information:

[Revision in the main text]

On page 8 Methods section, “After that, the rr-OHM film was fabricated using a method as described in reference [26]. 98% regioregular P3HT (molecular weight $M_w \sim 87,000$, Sigma Aldrich) 100 mg was dissolved in 1 ml chlorobenzene by heating the solution to 50 °C for 3 hours and resting at room temperature for 2 hours. Thin films were spin-coated on the AgNPs-PMMA layer at 5000 rpm for 60 s. Optical characterization of the OHM (film thickness: 182 nm) was performed via a rotating polarizer type spectroscopic ellipsometer (J.A. Woollam M-2000D, J. A. Wollam Co. Ltd.) as described in reference [26]. The optical transfer function calculation results show that the resolution improvement will be decreased with further increasing the thickness of the OHM film due to the absorption loss (Figure S8). Polystyrene beads (refractive index $n \sim 1.6$, radius $r = 40$ nm) are then drop-cast onto the OHM.”

[Revision in Supporting Information]

S8. The optical transfer function for OHM films with different thicknesses

Figure S8. Calculated optical transfer functions (OTFs) in the spatial frequency domain of 180-nm, 230-nm, and 480-nm thick OHM films.

We thank both reviewers again for their time and efforts on the careful examination of our manuscript. We hope the revisions we made in the manuscript have addressed all the review’s comments, and the manuscript in its revised form is considered suitable for publication in *Nature Communications*.

REVIEWER COMMENTS

Reviewer #1 (Remarks to the Author):

The authors have addressed all my concerns, so I recommend acceptance of the revised manuscript.

Reviewer #2 (Remarks to the Author):

In the revised manuscript, authors has addressed the concerns by providing a comparison of present work with other similar prior art on label-free super-resolution imaging. This comparison is based on resolution enhancement supported by different methods, which is a reasonable.

Application of Blind-SIM or other similar reconstruction algorithm on coherent scattering image could lead to reconstruction artefacts. For example, Kai Wicker and Rainer Heintzmann's commentary paper in Nature Photonics "Resolving a misconception about structured illumination", has provided discussion about the pitfalls of applying SIM method for coherent imaging (as in present case) and artificial narrowing of the PSF.

Wicker, K., Heintzmann, R. Resolving a misconception about structured illumination. Nature Photon 8, 342–344 (2014). <https://doi.org/10.1038/nphoton.2014.88>

Thus, it would have been beneficial to have a theoretical framework about enhancing the resolution in coherent imaging via Blind SIM and discussion around possible reconstruction artefacts for coherent imaging case. To this end performing imaging of real biological samples is useful instead of small nanobeads to determine such issues.

Reviewer #3 (Remarks to the Author):

Both referees of this manuscript [Yeon Lee et al., "Hyperbolic material enhanced scattering nanoscopy for label-free super resolution imaging"] provided an in-depth review of this work. All critical comments brought by both referees are adequately addressed in the authors' point-by-point response. The critical comment about the novelty of this work stands out in the report of the second referee and I would like to address it in my review. My general recommendation is that this manuscript can be published in Nature Communications after minor revision following my comments below.

1. Novelty: In a response letter, the authors stated that the novelty of this work is due to "...the fact that this is the first time to introduce OHM with optical properties much better than traditional optical materials." It is also in "...the record high resolution improvement." The comparative analysis of the resolution values in different papers presented as Table in the authors' reply is convincing. It resolves the issue and provides a clarification about the novelty of this work. I do believe, however, that some part of this comparative resolution analysis needs to be included in Introduction. Also, the papers mentioned by the second referee should be added to the list of references and discussed in Introduction. The wider point of the authors is that the 3-fold resolution improvement observed previously can be related to the role of high-index substrate whereas the 5.5-fold resolution

improvement observed in this work calls for novel mechanisms related to the use of hyperbolic materials.

2. Recently an article was published in Optics Express which is directly relevant to the resolution issues discussed above: J. Haug et al., “Confined hyperbolic metasurface modes for structured illumination microscopy,” Opt. Express 29, 42331 (2021). This paper uses hyperbolic materials and a similar blind SIM algorithm. It was submitted after this manuscript (on 8 September 2021) and the resolution improvement is only 3.1-fold compared to 5.5-fold demonstrated in this manuscript. It seems that this paper should be discussed in this manuscript.

3. In Introduction, after “Various label-free super-resolution imaging methods have recently been proposed” the authors can provide a reference source on this subject: “Label-Free Super-Resolution Microscopy”, V. Astratov (Ed.), Springer, Cham, Switzerland (2019).

4. Since the resolution enhancement is one of the main issues in this work, it creates a broader question if the resolution can be further increased by using contact microspheres for a label-free [Ann. Phys. (Berlin) 527, 513–522 (2015); Opt. Express 23, 24484-24496 (2015)] or FL [Nanoscale 9, 14907 (2017); Appl. Phys. Lett. 114, 131101 (2019)] imaging in combination with hyperbolic materials. Can the authors comment on this possibility?

We are grateful to all the reviewers for their positive evaluation of our work and for their valuable comments on this manuscript. A point-by-point response to these comments with the corresponding manuscript modifications is provided below, which is rendered in a different text format for clarity:

Blue Italic: Reviewer's comment;
Black Regular: Our response;
Red Regular: Modifications of the manuscript.

Response to Reviewer #1

The authors have addressed all my concerns, so I recommend acceptance of the revised manuscript.

We appreciate all the thoughtful comments and constructive suggestions from the reviewer which helped us improve the quality of the manuscript. And, we thank again the reviewer for his/her recommendation for publication.

Response to Reviewer #2

In the revised manuscript, authors has addressed the concerns by providing a comparison of present work with other similar prior art on label-free super-resolution imaging. This comparison is based on resolution enhancement supported by different methods, which is a reasonable.

We are glad to see that the reviewer satisfies the response to his/her main concern raised in the first round of revision. We thank the reviewer for his/her second revision on our manuscript, our response is provided below.

[Question 2-1] *Application of Blind-SIM or other similar reconstruction algorithm on coherent scattering image could lead to reconstruction artefacts. For example, Kai Wicker and Rainer Heintzmann's commentary paper in Nature Photonics "Resolving a misconception about structured illumination", has provided discussion about the pitfalls of applying SIM method for coherent imaging (as in present case) and artificial narrowing of the PSF.*

Wicker, K., Heintzmann, R. Resolving a misconception about structured illumination. Nature Photon 8, 342–344 (2014). <https://doi.org/10.1038/nphoton.2014.88>

Thus, it would have been beneficial to have a theoretical framework about enhancing the resolution in coherent imaging via Blind SIM and discussion around possible reconstruction

artefacts for coherent imaging case. To this end performing imaging of real biological samples is useful instead of small nanobeads to determine such issues.

[Reply 2-1] We thank the reviewer for the comment and suggestion. We have carefully checked out the reference that the reviewer provided, and we have realized that the enhancement mechanism behind our approach can easily be confused with that of structured illumination; this confusion gets even worse when different imaging modalities are involved.

We have thus presented a theoretical framework (as suggested by the reviewer) into a new supplementary section (i.e. Supplementary Section S8) in the revised Supporting Information, and then added a discussion on the resolution enhancement mechanism based on the theoretical framework in the revised manuscript, as follows:

[Revision in Supporting Information]

The fundamental principle of structured illumination microscopy (SIM) revolves around shifting high spatial frequency components of an object into the detectable bandwidth of far-field optics to achieve a larger effective imaging bandwidth. Here we provide a simple theoretical framework, adapted from Refs. [1] and [2], to describe the resolution enhancement via SIM in different imaging scenarios.

Since a fluorescence process responds to the illumination intensity while a scattering process responds to the illumination field, the detected fluorescence intensity $D(\mathbf{r})$ and the scattering intensity $d(\mathbf{r})$ of a sample $s(\mathbf{r})$ are

$$D(\mathbf{r}) = [I(\mathbf{r})s(\mathbf{r})] \otimes h(\mathbf{r}) \text{ and} \quad (1)$$

$$d(\mathbf{r}) = |[E(\mathbf{r})s(\mathbf{r})] \otimes a(\mathbf{r})|^2, \quad (2)$$

in the incoherent imaging and coherent imaging modalities, respectively. Here $I(\mathbf{r})$ is the illumination intensity and $h(\mathbf{r})$ is the incoherent detection point spread function (PSF), while $E(\mathbf{r})$ is the illumination field and $a(\mathbf{r})$ is the coherent detection PSF.

In the case of fluorescence imaging, the optical transfer function (OTF) $\tilde{h}(\mathbf{k})$ has a cutoff frequency at $\tilde{h}_{\text{cutoff}} = 2\text{NA} \times k_0$, where NA is the numerical aperture of the detection optics and k_0 is the free space wavevector at the emission wavelength. For scattering imaging, the coherent transfer function (CTF) $\tilde{a}(\mathbf{k})$ has a cutoff frequency at $\tilde{a}_{\text{cutoff}} = \text{NA} \times k_0$.

For a given illumination field $E_p(\mathbf{r})$ bounded by a maximum spatial frequency k_p , the corresponding intensity profile $I_p(\mathbf{r}) = |E_p(\mathbf{r})|^2$ is bounded by $2k_p$. It can be readily seen that the maximum achievable cutoff frequency k_c for fluorescence and scattering SIM are

$$k_{c,\text{fluo}} = 2\text{NA} \times k_0 + 2k_p \text{ and} \quad (3)$$

$$k_{c,\text{scat}} = \text{NA} \times k_0 + k_p. \quad (4)$$

Since the far-field detection bandwidth cannot be improved further other than using high NA optics, creating illumination pattern with high spatial frequency k_p is the key to improving the resolution of SIM.

With this theoretical framework in place, the resolution enhancement mechanism in different scenarios (I–IV) from different combinations of imaging modalities (coherent or incoherent) and SIM mediums (conventional optics or high- k material) can be clarified below:

- I. **Scattering imaging using conventional optics.** In conventional SIM, a sinusoidal illumination $E_p(\mathbf{r}) = \cos(k_p \mathbf{r})$ is generated based on interference of incident beams from two opposite illumination angles, with a maximum wavevector $k_{p,\max} = \text{NA} \times k_0$ limited by the illumination optics. However, since the scattering process is linear to the electric field, the collected object information is no more than imaging using oblique illuminations at one angle and then the other [1]. Both methods achieve an extended spatial frequency bandwidth of $k_{c,\text{scat}} = 2\text{NA} \times k_0$ —the Abbe diffraction limit—as described by Eq. 4. It is worth noting that, although conventional SIM will not yield any additional information beyond that provided by oblique illuminations in scattering imaging, reconstruction algorithms are still required to obtain the final image from SIM sub-images.
- II. **Fluorescence imaging using conventional optics.** In contrast to the scattering process, a fluorescence process responds to the excitation intensity instead of the field distribution. The interference illumination pattern in fluorescence SIM has an intensity distribution of $I_p(\mathbf{r}) = |E_p(\mathbf{r})|^2 = \frac{1}{2} + \frac{1}{2} \cos(2k_p \mathbf{r})$ with $k_{p,\max} = \text{NA} \times k_{\text{exc}}$, where k_{exc} is the free space wavevector at the excitation wavelength. The improved cutoff resolution is therefore $k_{c,\text{fluo}} = 2\text{NA} \times k_0 + 2\text{NA} \times k_{\text{exc}}$ as described by Eq. 3. In the case when the Stokes shift of the fluorescence is small, $k_{c,\text{fluo}} \approx 4\text{NA} \times k_0$, which corresponds to the commonly acknowledged 2-fold resolution improvement of SIM over the Abbe diffraction limit of $2\text{NA} \times k_0$ [2].
- III. **Fluorescence imaging with high- k material.** Illumination patterns with spatial frequency components beyond the traditional diffraction-limit are required to further improve the resolution of SIM. By using materials that support a larger lateral wavevector, such as high refractive index waveguides or hyperbolic metamaterials, the maximum illumination wavevector k_{mat} is limited by the material and can be much larger than that supported by conventional optics. The resulting cutoff frequency is $k_{c,\text{fluo}} = 2\text{NA} \times k_0 + 2k_{\text{mat}}$, leading to a resolution enhancement of $1 + k_{\text{mat}}/\text{NA} \times k_0$.
- IV. **Scattering imaging with high- k material.** Although conventional SIM will not yield any additional information beyond that provided by oblique illuminations in scattering imaging as described in the scenario I, structured illumination with high- k materials in this scenario does lead to an extended imaging bandwidth of $k_{c,\text{scat}} = \text{NA} \times k_0 + k_{\text{mat}}$, with a resolution enhancement of $1/2 \left(1 + k_{\text{mat}}/\text{NA} \times k_0\right)$ compared to the Abbe diffraction limit. **It is the exceptionally large k_{mat} in the high- k materials that leads to a gain in information compared to the conventional resolution.** Reconstruction is requested by the structured illumination to resolve the final super-resolution image.

In brief, the resolution enhancement in different scenarios is summarized in the table below.

	Scattering imaging	Fluorescence imaging
Conventional SIM	I: 1	II: 2
High- k material SIM	IV: $1/2 \left(1 + k_{\text{mat}}/\text{NA} \times k_0\right)$	III: $1 + k_{\text{mat}}/\text{NA} \times k_0$

Table S1: Resolution enhancement compared to the Abbe diffraction limit of $2\text{NA} \times k_0$ in different scenarios (I–IV) from different combinations of imaging modalities (coherent or incoherent) and SIM mediums (conventional optics or high- k material).

It is now clear that our current work is in the scenario IV where we make use of high- k illuminations supported by low-loss organic hyperbolic metamaterials to enhance the resolution of scattering imaging. To retrieve the super-resolution information, Blind-SIM algorithm is used for image reconstruction. Note that, while Blind-SIM and other similar joint-deconvolution algorithms for SIM reconstruction can lead to imaging artefacts when directly applied to coherent scattering images, we mitigate the issue by averaging the scattering sub-frames in the azimuthal direction to generate effective incoherent images of the scatters while retaining all the embedded high spatial frequency information.

The resolution enhancement defined in this theoretical framework allows for a relatively fair comparison among different super-resolution methods in the same imaging modality. To this end, the specific experimental conditions of a super-resolution method, such as the lens NA, the wavelength used, and the applied material system must be provided in addition to the achieved absolute resolution. For example, **Table S2** summarizes these specifications of the few state-of-the-art label-free scattering imaging methods, and the corresponding resolution enhancements are thereby derived. As can be seen, compared to the absolute resolution, the resolution enhancement is a more direct and precise parameter to evaluate a super-resolution method. The super-resolution method with a 5.5-fold resolution enhancement demonstrated in our current work has thus pushed the resolution limit of label-free nanoscopies into a new level.

Ref. #	Optical medium	Lens NA, Wavelength, Diffraction-limited resolution	Absolute resolution	Resolution enhancement
Our work	Organic hyperbolic materials	0.6 NA, $\lambda = 532$ nm, 443 nm	80 nm	~5.5 fold
3	Scalable photonic chip with grating	1.1 NA, $\lambda = 660$ nm, 300 nm	93 nm	~3 fold
4	Fluorescent nanowire ring	0.85 NA, $\lambda = 520$ nm, 306 nm	140 nm	~2 fold

5	Fluorescent polymer films	0.85 NA, $\lambda = 532$ nm, 313 nm	149 nm	~2 fold
6	Si ₃ N ₄ waveguide	0.95 NA, $\lambda = 488, 561, 647$ nm, 257, 295, 341 nm	160 nm	<2 fold

Table S2: Resolution enhancement in selected label-free scattering imaging methods.

[Revision in the main text]

The first paragraph of Discussion: “Enhancing the resolution of scattering microscopies is the main task of this work, so to have a better understanding of the resolution enhancement mechanism behind the proposed HMES nanoscopy a theoretical framework adapted from Refs. [1] and [2] is provided in **Supplementary Section S8**. According to the theoretical framework, the resolution enhancement of scattering imaging is $1/2 \left(1 + k_{\text{mat}}/NA \times k_0\right)$, where k_{mat} and k_0 are the maximum spatial frequencies supported by the material used and free space, respectively (**Table S1**). Since high refractive-index materials support a large k_{mat} and thus give rise to a high resolution enhancement, they are the dominant factor of scattering image’s resolution; unfortunately, as a result of lack of a high refractive-index material, current scattering imaging techniques have only demonstrated a resolution enhancement of ~3 fold³⁻⁶ (**Table S2**), as mentioned above. In the HMES nanoscopy demonstrated in this work, the high- k low-loss hyperbolic material OHM (**Figure S9**) is used, so a 5.5-fold resolution enhancement is enabled in scattering microscopies.”.

Comment on reconstruction artifacts

Since the focus of this manuscript is the first experimental demonstration of super-resolution scattering imaging using high- k low-loss organic hyperbolic metamaterials for a record resolution enhancement, our efforts were made to ensure the accuracy of the resolution value by using single nanoparticle samples, while eliminating/identifying possible reconstruction artifacts especially those that produce a wrong resolution, such as fine pseudo-structures from reconstructed nonmodulated background/noise signal. So, reconstruction artifacts were only briefly mentioned.

A systematic study is required to deal with the reconstruction artifacts in a SIM to obtain a high-quality super-resolution image. This study is expected to include identifying artifacts, figuring out their causes, and, if possible, objectively correcting for some common ones, in various application scenarios, which call for a series of works. Such a study, even without using real biological samples, is out of the scope of our current proof-of-concept work. Nevertheless, the theoretical framework provided above might be used as a guidance to clarify possible sources that cause reconstruction artifacts, and thus benefiting the future systematic studies of the reconstruction artifacts.

In response to the comment and suggestion, we have added more comments on reconstruction artifacts into the revised manuscript, as follows:

[Revision in the main text]

The second paragraph of Discussion: “Artifacts in the final reconstructed image of HMES nanoscopy can be caused by several factors, such as directional scattering of densely packed objects, in addition to image reconstruction-related artifacts and lack of sufficient frames for reconstruction. **A systematic study to deal with the artifacts is thus required in the future for obtaining a high-quality reconstructed super-resolution image. Such a study is expected to include identifying artifacts, figuring out their causes, and, if possible, objectively correcting for some common ones, in various application scenarios⁷. The theoretical framework provided in **Supplementary Section S8** might be used as a guidance to clarify possible sources that cause the artifacts, and thus benefiting the future systematic study.**”.

Response to Reviewer #3

Both referees of this manuscript [Yeon Lee et al., “Hyperbolic material enhanced scattering nanoscopy for label-free super resolution imaging”] provided an in-depth review of this work. All critical comments brought by both referees are adequately addressed in the authors’ point-by-point response. The critical comment about the novelty of this work stands out in the report of the second referee and I would like to address it in my review. My general recommendation is that this manuscript can be published in Nature Communications after minor revision following my comments below.

We appreciate the reviewer’s evaluation on the first round of revision of our manuscript, and we agree with him/her that it was an in-depth review, which certainly helped us improving the clarity and readability of our manuscript. And, we thank the reviewer for his/her general recommendation for publication and providing us further comments, our point-by-point response to these comments is provided below.

[Question 3-1] *1. Novelty: In a response letter, the authors stated that the novelty of this work is due to “...the fact that this is the first time to introduce OHM with optical properties much better than traditional optical materials.” It is also in “...the record high resolution improvement.” The comparative analysis of the resolution values in different papers presented as Table in the authors’ reply is convincing. It resolves the issue and provides a clarification about the novelty of this work. I do believe, however, that some part of this comparative resolution analysis needs to be included in Introduction. Also, the papers mentioned by the second referee should be added to the list of references and discussed in Introduction.*

The wider point of the authors is that the 3-fold resolution improvement observed previously can be related to the role of high-index substrate whereas the 5.5-fold resolution improvement observed in this work calls for novel mechanisms related to the use of hyperbolic materials.

[Reply 3-1] We thank the reviewer for his/her agreement that the comparative analysis about

the resolution enhancement over different methods is convincing. We also agree with the reviewer that highlighting this comparative resolution analysis with all relative references provided by Reviewer #2 is valuable. So, first, we have presented a theoretical framework of resolution enhancement provided above into a new supplementary section (i.e. Supplementary Section S8) in the revised Supporting Information. Then, we have added more explanation about the limitation of resolution enhancement in Introduction of the revised manuscript, as follows:

[Revision in the main text]

The third paragraph of Introduction: “Nevertheless, in contrast to fluorescence super-resolution nanoscopies, the resolution of the current super-resolution scattering imaging techniques is limited by the substrate materials that support a high effective refractive index at the desired wavelength. On one hand, dielectric materials have a limited refractive index. For example, super-resolution scattering image with a ~3-fold resolution improvement has been obtained by using GaP⁸ and it is hard to find any other materials which are better than GaP. On the other hand, plasmonic materials have shown much higher effective refractive indices at visible frequencies; therefore, super-resolution fluorescence microscopies based on plasmonic materials ranging from surface plasmon polariton structures⁹⁻¹¹ to hyperbolic metamaterials have recently been theoretically proposed and experimentally demonstrated¹²⁻¹⁴. However, these plasmonic materials have only been applied to fluorescence imaging based on our knowledge. As a result, current label-free scattering imaging techniques have only demonstrated a resolution improvement of ~3 fold³⁻⁶.”.

Next, we have added a discussion on the resolution enhancement based on the theoretical framework in Discussion of the revised manuscript, as follows:

[Revision in the main text]

The first paragraph of Discussion: “Enhancing the resolution of scattering microscopies is the main task of this work, so to have a better understanding of the resolution enhancement mechanism behind the proposed HMES nanoscopy a theoretical framework adapted from Refs. [1] and [2] is provided in **Supplementary Section S8**. According to the theoretical framework, the resolution enhancement of scattering imaging is $1/2 \left(1 + k_{\text{mat}}/NA \times k_0 \right)$, where k_{mat} and k_0 are the maximum spatial frequencies supported by the material used and free space, respectively (**Table S1**). Since high refractive-index materials support a large k_{mat} and thus give rise to a high resolution enhancement, they are the dominant factor of scattering image’s resolution; unfortunately, as a result of lack of a high refractive-index material, current scattering imaging techniques have only demonstrated a resolution enhancement of ~3 fold³⁻⁶ (**Table S2**), as mentioned above. In the HMES nanoscopy demonstrated in this work, the high- k low-loss hyperbolic material OHM (**Figure S9**) is used, so a 5.5-fold resolution enhancement is enabled in scattering microscopies.”.

Lastly, we have added all relative papers mentioned by Reviewer #2 to the list of references.

We also appreciate the reviewer’s concise summary of resolution enhancement mechanism,

which is exactly in line with the theoretical framework we have provided above.

[Question 3-2] *2. Recently an article was published in Optics Express which is directly relevant to the resolution issues discussed above: J. Haug et al., “Confined hyperbolic metasurface modes for structured illumination microscopy,” Opt. Express 29, 42331 (2021). This paper uses hyperbolic materials and a similar blind SIM algorithm. It was submitted after this manuscript (on 8 September 2021) and the resolution improvement is only 3.1-fold compared to 5.5-fold demonstrated in this manuscript. It seems that this paper should be discussed in this manuscript.*

[Reply 3-2] We thank the reviewer for pointing this paper out. We have checked it out and found that this paper reported **numerical results on super-resolution fluorescence imaging** with a 3.1-fold resolution improvement by using Ag-based hyperbolic metamaterials, i.e. **in the scenario III** according to the theoretical framework provided above—please see the details in **[Reply 2-1]**. In our current manuscript, however, we **experimentally demonstrated super-resolution scattering imaging in the scenario IV** so we cannot make a comparison between our current manuscript with this paper.

We can make a comparison between this paper with our previous publications on fluorescence nanoscopies: 1. In the work [Nature Communications 12, 1559 (2021)], we reported **experimental results on super-resolution fluorescence imaging** with a 7-fold resolution improvement (520-nm emission, 0.6 NA, 60-nm resolution) by using Ag-based hyperbolic metamaterials; 2. In another work [Advanced Science 8, 2170149 (2021)], we used low-loss organic hyperbolic metamaterials to experimentally demonstrate **a super-resolution fluorescence imaging** with a 17-fold resolution enhancement (520-nm emission, 0.6 NA, 25-nm resolution).

In short, our current manuscript is the first experimental demonstration of **super-resolution scattering imaging** using (organic) hyperbolic metamaterials, so there are no other similar reports, so far as we know.

In response to the comment and suggestion, we have added a discussion about hyperbolic metamaterials-based super-resolution fluorescence imaging with these recent publications in the revised manuscript, as follows:

[Revision in the main text]

In the third paragraph of Introduction: “**On the other hand, plasmonic materials have shown much higher effective refractive indices at visible frequencies; therefore, super-resolution fluorescence microscopies based on plasmonic materials ranging from surface plasmon polariton structures⁹⁻¹¹ to hyperbolic metamaterials have recently been theoretically proposed and experimentally demonstrated¹²⁻¹⁴. However, these plasmonic materials have only been applied to fluorescence imaging based on our knowledge. As a result, current label-free scattering imaging techniques have only demonstrated a resolution improvement of ~3 fold³⁻⁶.”**

[Question 3-3] 3. *In Introduction, after “Various label-free super-resolution imaging methods have recently been proposed” the authors can provide a reference source on this subject: “Label-Free Super-Resolution Microscopy”, V. Astratov (Ed.), Springer, Cham, Switzerland (2019).*

[Reply 3-3] We thank the reviewer for pointing out this book, which is indeed relevant to our current manuscript. So, we have added it as a reference in Introduction of the revised manuscript.

[Revision in the main text]

First sentence of the second paragraph of Introduction: “Various label-free super-resolution imaging methods have recently been proposed¹⁵.”

[Question 3-4] 4. *Since the resolution enhancement is one of the main issues in this work, it creates a broader question if the resolution can be further increased by using contact microspheres for a label-free [Ann. Phys. (Berlin) 527, 513–522 (2015); Opt. Express 23, 24484-24496 (2015)] or FL [Nanoscale 9, 14907 (2017); Appl. Phys. Lett. 114, 131101 (2019)] imaging in combination with hyperbolic materials. Can the authors comment on this possibility?*

[Reply 3-4] We thank the reviewer for the comment and suggestion. From the resolution enhancement point of view, we have no doubt that this possibility exists. Actually, one of the references that the reviewer mentioned, i.e. the reference [Nanoscale 9, 14907 (2017)], is from our group, so, we were thinking of this possibility to push further the resolution limit.

We can simply think of a contact microsphere as a solid immersion lens, which gives rise to an increased detection NA so that both the incoherent and coherent detection PSFs in Eqs. 1 and 2 will be smaller, leading to an extra resolution enhancement for both the label-free or FL imaging cases. Moreover, we believe that coupling whispering-gallery modes in microsphere with hyperbolic modes in hyperbolic materials will further improve the resolution.

In response to the comment and suggestion, we have added a note about further resolution enhancement by combining contact microspheres with hyperbolic metamaterials in the revised manuscript, as follows:

[Revision in the main text]

Last sentence of the first paragraph of Discussion: “**It is also worth pointing out that the resolution enhancement for both the high- k hyperbolic metamaterials-based scattering or fluorescence nanoscopies could be further pushed with an increased detection NA, such as using a contact microsphere^{16–19} or coupling whispering-gallery modes in the microsphere²⁰.**”.

We thank all reviewers again for their time and efforts on the careful examination of our manuscript. We hope the revisions we made in the manuscript have addressed all the reviewer's comments, and the manuscript in its revised form is considered suitable for publication in *Nature Communications*.

References

1. Wicker, K. & Heintzmann, R. Resolving a misconception about structured illumination. *Nat. Photonics* **8**, 342–344 (2014).
2. Aaminski, C. L. F. K. *et al.* Frontiers in structured illumination microscopy. *Optica* **3**, 667 (2016).
3. Ventalon, C. & Mertz, J. Quasi-confocal fluorescence sectioning with dynamic speckle illumination. *Opt. Lett.* **30**, 3350 (2005).
4. Kim, M. K., Park, C. H., Rodriguez, C., Park, Y. K. & Cho, Y. H. Superresolution imaging with optical fluctuation using speckle patterns illumination. *Sci. Rep.* **5**, 1–10 (2015).
5. Guo, K. *et al.* 13-Fold Resolution Gain Through Turbid Layer Via Translated Unknown Speckle Illumination. *Biomed. Opt. Express* **9**, 260 (2018).
6. Diekmann, R. *et al.* Chip-based wide field-of-view nanoscopy. *Nat. Photonics* **11**, 322–328 (2017).
7. Demmerle, J. *et al.* Strategic and practical guidelines for successful structured illumination microscopy. *Nat. Protoc.* **12**, 988–1010 (2017).
8. van Putten, E. G. *et al.* Scattering Lens Resolves Sub-100 nm Structures with Visible Light. *Phys. Rev. Lett.* **106**, 193905 (2011).
9. Wei, F. *et al.* Wide Field Super-Resolution Surface Imaging through Plasmonic Structured Illumination Microscopy. *Nano Lett.* **14**, 4634–4639 (2014).
10. Bezryadina, A., Zhao, J., Xia, Y., Zhang, X. & Liu, Z. High Spatiotemporal Resolution Imaging with Localized Plasmonic Structured Illumination Microscopy. *ACS Nano* **12**, 8248–8254 (2018).
11. Bezryadina, A. *et al.* Localized plasmonic structured illumination microscopy with gaps in spatial frequencies. *Opt. Lett.* **44**, 2915 (2019).
12. Lee, Y. U. *et al.* Metamaterial assisted illumination nanoscopy via random super-resolution speckles. *Nat. Commun.* **12**, 1559 (2021).
13. Lee, Y. U. *et al.* Organic Hyperbolic Material Assisted Illumination Nanoscopy. *Adv. Sci.* **8**, 2102230 (2021).
14. Haug, J. *et al.* Confined hyperbolic metasurface modes for structured illumination microscopy. *Opt. Express* **29**, 42331 (2021).
15. Astratov, V. *Label-Free Super-Resolution Microscopy*. (Springer International Publishing, 2019). doi:10.1007/978-3-030-21722-8.
16. Allen, K. W. *et al.* Super-resolution microscopy by movable thin-films with embedded

- microspheres: Resolution analysis. *Ann. Phys.* **527**, 513–522 (2015).
17. Allen, K. W. *et al.* Overcoming the diffraction limit of imaging nanoplasmonic arrays by microspheres and microfibers. *Opt. Express* **23**, 24484 (2015).
 18. Bezryadina, A. *et al.* Localized plasmonic structured illumination microscopy with an optically trapped microlens. *Nanoscale* **9**, 14907–14912 (2017).
 19. Brettin, A. *et al.* Enhancement of resolution in microspherical nanoscopy by coupling of fluorescent objects to plasmonic metasurfaces. *Appl. Phys. Lett.* **114**, 131101 (2019).
 20. Vahala, K. J. Optical microcavities. *Nature* **424**, 839–846 (2003).

REVIEWERS' COMMENTS

Reviewer #3 (Remarks to the Author):

The authors have addressed all my concerns, so I recommend that the revised manuscript should be accepted for publication in Nature Communications.